# Links between seasonal suprapermafrost groundwater, the hydrothermal change of the active layer, and river runoff in alpine permafrost watersheds

Jia Qin[1,2], Yongjian Ding[1,2*], Faxiang Shi[1,2], Junhao Cui[1,2], Yaping Chang[1,3], Tianding Han[1,2], Qiudong Zhao[2,3]

[1] State Key Laboratory of Cryospheric Sciences, Northwest Institute of Eco-Environment and Resources, Chinese Academy of Sciences, Lanzhou, 730000, China
[2] University of Chinese Academy of Sciences, Beijing, 100049, China
[3] Key Laboratory of Eco-hydrology Inland River Basin, Northwest Institute of Eco-Environment and Resources, Chinese Academy of Sciences, Lanzhou, 730000, China

*Correspondence to*: Yongjian Ding (dyj@lzb.ac.cn)

**Abstract.** The seasonal dynamic of the suprapermafrost groundwater significantly affects the runoff generation and confluence in permafrost basins and is a leading issue that must urgently be addressed in hydrological research in cold and alpine regions. In this study, the seasonal dynamic process of the suprapermafrost groundwater level (SGL), vertical gradient changes of soil temperature (ST), moisture content in the active layer (AL), and river level changes were analyzed at four permafrost watersheds in the Qinghai–Tibet Plateau using comparative analysis and the nonlinear correlation evaluation method. The impact of freeze–thaw processes on seasonal SGL and the links between SGL and surface runoff were also investigated. The SGL process in a hydrological year can be divided into four periods: (A) a rapid falling period (October–middle November), (B) a stable low-water period (late November–May), (C) a rapid rising period (approximately June), and (D) a stable high-water period (July–September), which synchronously respond to seasonal variations in soil moisture and temperature in the AL. The characteristics and causes of SGL changes significantly varied during these four periods. The freeze–thaw process of the AL regulated SGL and surface runoff in permafrost watersheds. During period A, with rapid AL freezing, the ST had a dominant impact on the SGL; in period B, the AL was entirely frozen due to the stably low ST, while the SGL dropped to the lowest level with small changes. During period C, ST in the deep soil layers of AL (below 50 cm depth) significantly impacted the SGL (nonlinear correlation coefficient $R^2 > 0.74$, $P < 0.05$), whereas the SGL change in the shallow soil layer (0－50 cm depth) showed a closer association with soil moisture content. Rainfall was the major cause for the stable high SGL during period D. In addition, the SGLs in periods C and D were closely linked to the retreat and flood processes of river runoff. The SGL contributed approximately 57.0–65.8% of the river runoff changes in period D. These findings will help to facilitate future hydrological research in the permafrost basins and the development and utilization of water resources in cold and alpine regions.

# 1 Introduction

The groundwater in permafrost watersheds comprises suprapermafrost, intrapermafrost, and subpermafrost groundwater (Clark et al., 2001; Mavromatis et al., 2014; Huang et al., 2020). The suprapermafrost groundwater (SG) refers to the groundwater distributed above the permafrost layer; its stable floor is the permafrost table, which is primarily replenished by rainfall, surface water, and lateral flow in the active layer (AL) (Ma et al., 2017; Tregubov et al., 2022). When the surface soil is frozen, most surface replenishment sources of the SG are cut off, whereas, during the summer thawing season, the SG becomes non-confined water with a free surface (Qin et al., 2022). SG significantly impacts the regional water cycle and the supply-demand relationship of ecological water in permafrost watersheds (Huang et al., 2020; Gao et al., 2021) and plays a crucial role in regulating land surface processes and hydrology in cold regions (Wellman et al., 2013; Chang et al., 2015; Liu et al., 2021).

The suprapermafrost groundwater level (SGL) maintains a high value during the summer half of the year due to the quantity of rainfall and surface water frequently infiltrating into the thawed AL (Tregubov et al., 2021; Wei et al., 2021). The dynamics of the SGL are directly affected by rainfall and the surface meltwater supply (Young et al., 2000). Due to the impermeability of the permafrost table, SG can flow out of low-lying areas after reaching a particular level or laterally supply river runoff or lakes (Krickov et al., 2018; O'Neill et al., 2020; Gao et al., 2021; Qin et al., 2022). In the winter half of the year, with decreasing air temperature and surface soil freezing, most SG transforms into ground ice stored in AL (Xu et al., 2021). SG significantly impacts hydrological processes and water cycles in the permafrost basins through water migration and transformation (Ge et al., 2011; Chen et al., 2018). SG is one of the primary water sources for lake and river runoffs in a permafrost basin, especially during the summer AL thawing period. In some continuous permafrost basins, such as the source area of the Shule River in the northeastern part of the Qinghai–Tibet Plateau (QTP), SG contributes to over 30% of the total river runoff (Qin et al., 2022). The SG replenished over 60 mm of water into the Thermokarst Lake in the Beiluhe watershed of the QTP from June 20 to October 26, 2019, when the surface runoff flow into the Thermokarst Lake was only approximately 170 mm (Gao et al., 2021). The effect of frozen SG in the winter should be considered since the ice stored in AL could rapidly thaw during spring and supply a substantial amount of water to the spring flood. For example, at least one-third of the Shestakovka River spring flood is attributed to melted superpermaforst ice (Lebedeva, 2019), and the measured SG contributes to over 60% of the total discharge in the Ugol'naya-Dionisiya River at the beginning of the warm season (Tregubov et al., 2021). In addition, SG is a major source of baseflow in cold river basins. The SG contribution to base flow is over 90% in the continuous permafrost region of the Yukon River Basin around the Arctic (Walvoord et al., 2012).

Permafrost thaw is closely linked to soil moisture and temperature (Schuur and Abbott, 2011). The dynamics of SGL are closely associated with seasonal hydrothermal changes and freeze–thaw processes of AL. The SGL has a correspondingly distinct response to the freezing–thawing–refreezing of AL (Renzheng and Juan, 2019), a significant hydrological characteristic that differs from that in non-frozen soil regions (Wei et al., 2018). Previous studies on seasonal SG have primarily focused on the characteristics of SGL change in different freeze–thaw stages of AL in basins of the high

latitudes of the Northern Hemisphere and QTP (Chang et al., 2015; Throckmorton et al., 2016; O'Connor et al., 2019; Wei et al., 2021) and the impact factors of SG variation, including the climate (such as rainfall and air temperature (Dugan et al., 2009; Zhang et al., 2021), geological conditions (Woo and Xia, 1995; Sjöberg et al., 2013), soil properties (Raudina et al., 2018) and vegetation types (Koch et al., 2022), as well as the slope and aspect of the permafrost watershed (Tregubov et al., 2021; Wei et al., 2021). O'Connor et al. (2019) have previously reported that the SGL during early summer (June) was lower than that during late summer (August) in an Arctic watershed. Chang et al. (2015) and Gao et al. (2021) reported that SGL significantly rose during the AL thawing period during summer, and rapidly fell following land surface freezing in autumn and early-winter in the source region of the Yangtze River. These studies recognized the significant impact of seasonal SG on the ecology and interaction between surface water and groundwater in cold regions, which further highlighted the necessity of systematic investigations into the seasonal changes of SG in the Qinghai Tibet Plateau, as an important cold region.

Changes in air temperature directly affect the thickness and the freeze-thaw process of AL, as well as the water-resisting effect of permafrost (Chang et al., 2015). That further initiates the replenishment process and dynamic changes of SGL (O'Connor et al., 2019; Wei et al., 2021). Precipitation is the primary water source of SG. Especially in the thawing period, rainfall dominates the SGL variation (Dugan et al., 2009; Zhang et al., 2021). In addition, the SGL varies in permafrost regions with different vegetation and soils due to differences in migration rates and infiltration amounts of surface water to SG. For example, the hydrograph of SGL in alpine meadows significantly differs from that of alpine grasslands and bare land during the same rainfall events. In non-vegetation regions, SGL peaks and rapidly responds to rainfall (Koch et al., 2022). In addition, the SG depth varies in different land cover of specific regions; for instance, the SG depth in the swamp meadow of the Yellow River source is 0.1–0.8 meters, 0.8–2 meters in the alpine grassland and 2–8 meters in the desertification grassland (Wenbing et al., 2003). Moreover, significant differences have been reported in the interactive transport between SG and river runoff under different topographies, especially under different slope aspects. Wei et al. (2018, 2021) observed a lower terrain migration trend for SG during the thawing season and proposed that SG flows into and replenish nearby rivers or Thermokarst Lakes. Renzheng (2019) reported that a small amount of water could infiltrate AL during the initial freeze period of surface soil, thereby lowering SGL compared with river levels (RL); therefore, SG would be resupplied by nearby rivers. However, these findings lack supporting analysis based on field observation data. It is therefore necessary to conduct a detailed study on the "SG-RL" dynamic to clarify its linkage to topography changes.

Data from previous studies have expanded our understanding of SG, the effect of which has improved the development of permafrost hydrology. It is necessary to systematically investigate the linkage between seasonal hydrothermal changes of AL, SG, and surface runoff. This unclear linkage, which has been regarded as a "black box" in hydrological analyses and simulations (Yongjian et al., 2017), is a bottleneck problem in permafrost hydrological studies (Ge et al., 2011; Lafrenière et al., 2019). Identifying seasonal variations in SG and its hydrological linkages based on systematic observations is essential in cold regions, especially in the context of climate warming.

To better understand the dynamic rules and driving factors of SG, this study selected continuous permafrost watersheds in different locations of the QTP to address two challenges: (1) the seasonal dynamic pattern and spatial differentiation law of SG and (2) the influence of the AL freeze–thaw process on the dynamic changes in SG and river runoff in permafrost watersheds. Based on field observations, this study revealed the seasonal variation patterns of SG to provide theoretical support and facilitate regional water cycle research in permafrost regions.

## 2 Study area and materials

### 2.1 Selected research stations

The long-term cold climate and special geological tectonic movements contribute to the special development process and distribution characteristics of the permafrost in QTP (Cheng et al., 2019). With the rising temperatures, the permafrost area in the plateau has decreased to 1.06 million $km^2$ (or 40% of the total area of the QTP), yet remains the largest and highest permafrost region in the middle and low latitudes (Zhao et al., 2020). The climate in the QTP has become warmer and wetter, with the warming rate almost twice the global average (Zhao et al., 2020). With permafrost degradation, the permafrost has become thinner and some island permafrost has also begun to thaw, making groundwater dynamic gradually active. There is a large amount of groundwater in the QTP, while several areas are affected by permafrost and AL change. A large proportion of groundwater burial is shallow, and the thickness of the aquifer is relatively thin (<3 m) (Cheng and Jin, 2013).

Four permafrost stations were selected in this study, namely HLG, SL, MD, and FHS, at different locations on the QTP (Fig. 1) for comparative analysis of the meteorological and hydrothermal changes in AL and river runoff. The HLG and SL experimental stations are located in the headwater regions of the Heihe and Shule Rivers in the Qilian Mountains of the northeastern QTP, respectively, while the MD and FHS stations are in the headwater regions of the Yellow and Yangtze Rivers in the hinterland of the QTP, respectively. The mean annual precipitation at these four stations ranges from 100–400 mm. Stations HLG and SL in the Qilian Mountains have higher annual precipitation than stations FHS and MD. The annual precipitation at the HLG station is the highest (403.4 mm), whereas that at the MD station is 133.7 mm. The mean annual temperature in the four stations ranges from –5.2 °C (FHS) to 2 °C (MD), while the ground temperature ranges from –1.7 °C (FHS) to 0.3 °C (HLG). The maximum AL thawing depth at the research sites changes from 1.5 m at the FHS observation station to 2.5 m at the HLG station in summer. The primary vegetation types of the stations are alpine meadows, alpine grasslands, and swamp meadows (Table 1).

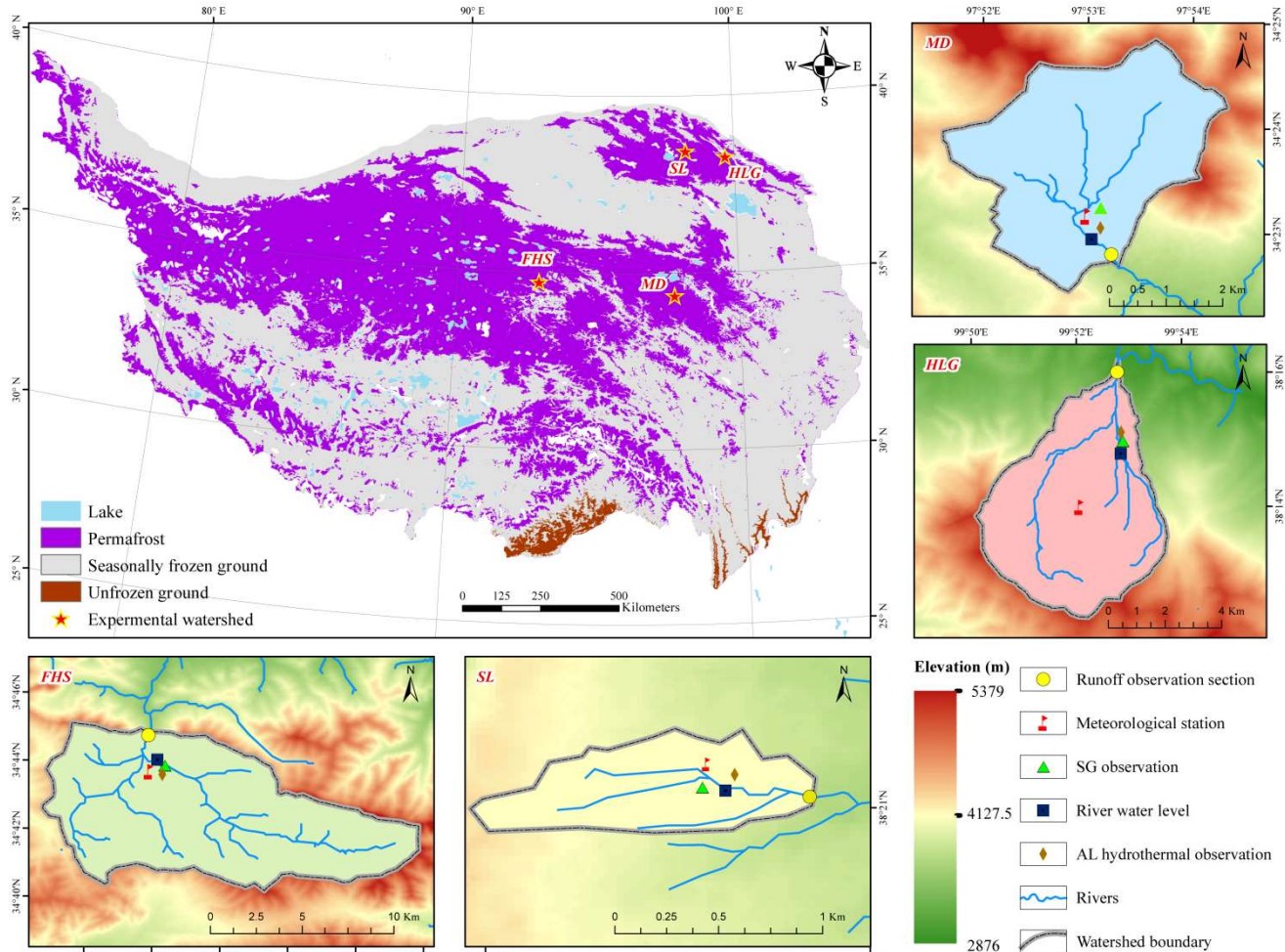

**Figure 1: Permafrost distribution and locations of the four SG observation stations in the QTP.**

125

**Table 1. The parameter characteristics in the four experimental watersheds of the Qinghai-Tibet plateau.**

|  | Longitudes | Latitudes | P (mm) | Ta (℃) | Tg (℃) | Hmax (m) | vegetation type |
|---|---|---|---|---|---|---|---|
| **HLG** | 99°50′-99°54′ E | 38°12′-38°16′ N | 403.4 | -3.1 | -0.3 | 2.5 | Alpine grassland |
| **SL** | 98°16′-98°17′ E | 38°20′-38°21′ N | 350.2 | -3.5 | -0.5 | 2.0 | Alpine meadow |
| **MD** | 97°52′-97°54′ E | 34°22′-34°25′ N | 133.7 | -2.0 | -1.0 | 1.8 | Swamp meadow |
| **FHS** | 92°51′-93°20′ E | 34°40′-34°46′ N | 290.9 | -5.2 | -1.7 | 1.5 | Alpine meadow |

Note: P is the annual precipitation, Ta is the annual mean air temperature, Tg is the annual mean ground temperature, and Hmax is the maximum yearly depth of AL.

130

The AL thickness of the four experimental watersheds is influenced by vegetation coverage and slope orientation. In areas with poor vegetation coverage and sunny slopes, the total thickness of the permafrost layer does not exceed 100 m; conversely, in areas with well-developed vegetation and shady slopes, the maximum thickness of the permafrost layer can reach 300 m, while the permafrost layer on the riverbed is the thinnest (lower than 50 m in FHS). In addition, AL and deeper permafrost layers contain abundant ground ice, the thawing and freezing of which can significantly affect SVMC change. The selected watersheds are commonly located in relatively open terrain with wide and shallow valleys, with mean annual river discharge of $0.12 \times 10^8$ km$^3$ (HLG), $0.2 \times 10^8$ km$^3$ (FHS), $0.4 \times 10^8$ km$^3$ (MD), and $1.0 \times 10^8$ km$^3$ (SL). Over 80% of the RL is concentrated in the summer. Storage time of groundwater, especially SG, in the experimental watersheds is relatively short and relies heavily on the supply of precipitation. SG is discharged by converging into adjacent rivers. The active period of groundwater is June–October; the SG around the river has been stored for a long time. Affected by the river water, there remains a high SG level in November, with sufficient replenishment and more active movement of groundwater. During summer, groundwater replenishes the river water, while in autumn and early winter, the RL is above the SGL, leading to the replenishment of SG from river water.

## 2.2 Experimental data

This study was primarily based on soil temperature (ST) and soil volumetric moisture content (SVMC) measured at different depths of AL, as well as the SGL data at the four permafrost stations (Fig. 1). The analysis of ST, SVMC, and SGL was conducted in a hydrological year (from 1$^{st}$ of October to 30$^{th}$ of September) and the time scale was daily. The data were obtained from the field observation stations of the Chinese Academy of Sciences, along with existing literature (Chang et al., 2015; Wei et al., 2018; Qin et al., 2022). The daily data of RL during the wet season (in summer) and water recession period (in autumn) in SL and FHS watersheds was also measured and compared with SGL to analyze the "SGL-RL" dynamic relationship. Daily rainfall data for the same hydrological year were obtained from automatic rain and snow gauges (T-200B) (30 min) at the permafrost stations and national weather stations (http://data.cma.cn/) in the study areas.

## 2.3 Observation of soil hydrothermal change and SGL

ST and SVMC at different AL depths were the two major soil hydrothermal parameters used in this study. In SL and HLG stations, ST and SVMC were continuously measured using HydraProbe Lite sensors with an accuracy of 0.3 °C and ±1.0%. The ST in the MD and FHS stations were measured using S-TMB-M006 sensors with an accuracy of 0.2 °C and the SVMC was measured using S-SMC-M005 sensors with an accuracy of ±3.0%. All sensors are suitable for use at –40–50 °C. All probes used for measuring the ST and SVMC were buried in the soil from the ground surface (10 cm depth) to the permafrost table. Probes were installed in the soil at depths of 10, 20, 40, 60, 80, 120, and 160 cm. All instruments were attached to a CR1000 data logger for data acquisition at each station, and recorded data were collected every 30 min. The ST and SVMC data in the SL, MD, and FHS stations were used to analyze the hydrothermal change of AL in a hydrological year since some data of ST and SVMC in the HLG station was missing due to instrument damage.

The SGLs at the four stations were measured using a HOBO pressure water level logger (U 20-001-04) produced in the United States. The logger, a built-in pressure water level sensor with a fully enclosed titanium alloy shell, is suitable for the automatic observation of groundwater in alpine environments. The measurement accuracy was high, with a resolution of 0.014 kPa (0.14 cm water depth). A water level logger was set in the AL groundwater wells at the four stations; the HOBOware Pro software was used to operate the logger. Using a reference water level, HOBOware Pro automatically converts pressure readings into water level readings, providing SGL data for a specific measurement period. In addition, some data from manually measuring the groundwater level with a ruler were used to compare and correct the HOBO-observed SGL data in different wet and dry periods. Daily SGL was calculated by averaging all corrected water level data obtained every 30 min daily.

The SGL, SVMC, and ST sensors were calibrated prior to field observations, which included measuring the initial value of the sensors, recording the measurement results, and adjusting the measurement results to achieve accurate values. We excavated the original soil in AL layer by layer and buried the probes; undisturbed soils were backfilled in different layers. During the initial days of the SGL, SVMC, and ST sensor function, the data were influenced by the unstable soil structure and were therefore excluded from analysis. Every 30 min recorded data was assessed to eliminate abnormally high or low values, which were interpolated by adjacent data. In addition, typical alpine hillslopes were selected in the central part of the SL and FHS experimental watersheds where the groundwater flow field on the hillslopes was cut by the river (with ground ice exposed at the edge of the riverbed with an obvious exchange between SG and river runoff), to observe the "SGL-RL" linkage during the AL thawing period.

## 2.4 Correlation analysis between the SGL and soil hydrothermal parameters in the AL

To better investigate seasonal SGL, the tipping points of the SGL data series were analyzed using the Pettitt test (Pettitt, 1979). A contour map was created using SigmaPlot (SigmaPlot 14.0, 2020) software to analyze the ST and SVMC changes at different depths of AL. To analyze the impact of soil hydrothermal parameters on SGL, the Boltzmann formula was used to perform nonlinear fitting between ST and SGL as well as SVMC and SGL at different AL depths. The formula was optimum for nonlinear fitting analysis between SG and hydrothermal variables in AL (Wang et al., 2012; Chang et al., 2015), as confirmed using the Levenberg–Marquardt method (Bates and Watts, 1988) and Universal Global Optimization algorithm (Benson, 2002). SPSS (SPSS 18.0, 2016) was used to perform nonlinear correlation analysis. The fitting and correlations were evaluated using the coefficient of determination ($R^2$) and root mean square error (RMSE). The Boltzmann formula is expressed as follows:

$$H = H_0 + \frac{a}{1 + e^{-\left(\frac{T - T_0}{b}\right)}}$$

where H is the SGL, and $T$ and $T_0$ are the ST of the target depths and the initial ST during the calculation period at that depth, respectively (unit: °C); $H_0$ is the initial SGL, and $a$ and $b$ are undetermined parameters associated with soil characteristics at different depths.

### 3  Seasonal characteristics of SGL, ST, and SVMC

As shown in Fig. 2, the SG data series during the hydrological year have four tipping points in each station. The Pettitt test results indicated that the trend of the streamflow series changed on 4[th] of December, 11[th] of June, and 5[th] of July in the HLG station; 17[th] of November, 10[th], and 25[th] of June, in the SL station; 3[rd] of November, 5[th] of December, 6[th] of June, and 10[th] of July in the MD station; and 29[th] of October, 20[th] of November, 7[th], and 26[th] of June in the FHS station. According to the tip points, which refer to the start or end times of different periods, there are four typical periods of SGL variation throughout the hydrological year at different stations on the QTP, which can be divided as follows: (A) rapid falling period, (B) stable low-water period, (C) rapid rising period, and (D) stable high-water period. Periods A–D began in mid-autumn (October–early November), early winter (late November–early December), early summer (June), and mid-summer (July), respectively. The specific start and end time of each period, as well as the duration of each period, showed minimal difference among the four stations (Fig. 2).

The ST and SVMC in the AL showed synchronous responses to seasonal variations in SGL (Figs. 2 and 3). ST rapidly decreased to 0 ℃ (rapid freezing) and stable low temperature below 0 ℃ (frozen stability), rapidly increased above 0 ℃ (rapid thawing) and fluctuated above 0 ℃ (thawing stability) in Periods A, B, C, and D, respectively. The SVMC also has four corresponding stages, namely rapid reduction, stable low value, rapid rise and fluctuation with the complete melting of the AL.

To some extent, SVMC could be a dynamic indicator of SGL. According to the vertical variations in seasonal SVMC and SGL at the experimental sites (Fig. 3), summer SGL fluctuates at depths where SVMC has a high value. For example, summer SGL at the SL site primarily fluctuated at 40–60 cm depths, where the SVMC had a high value, while the soil remained saturated or nearly saturated for an extended duration. Likewise, the summer SGL at the MD station fluctuated in 10–80 cm depths, where the SVMC also maintained a higher value (Fig. 2). Exploring the soil characteristics and SVMC in the AL can clarify the SGL dynamics in a specific area.

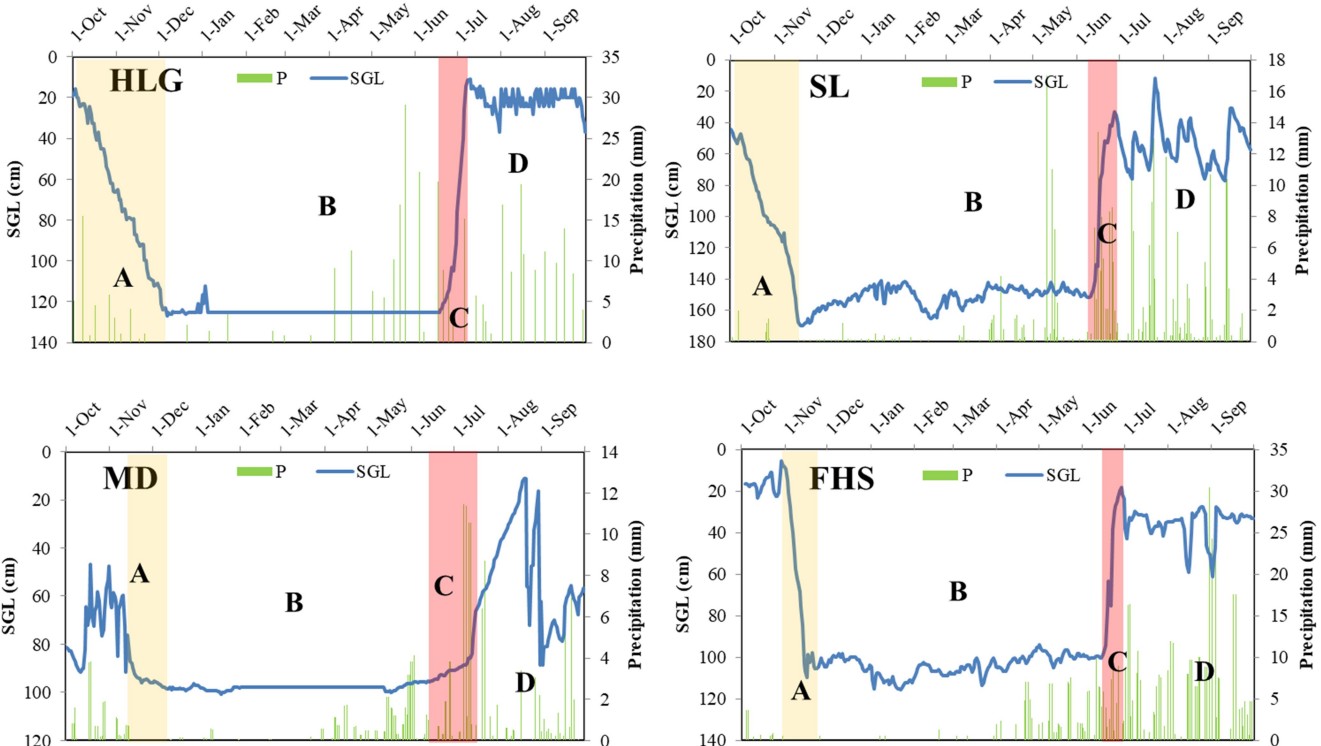

Figure 2: Four major phases of the seasonal suprapermafrost groundwater level (SGL) and the corresponding daily precipitation (P) in different sites of the QTP. A–D refer to the rapid falling period, stable low-water period, rapid rising period and stable high-water period, respectively.

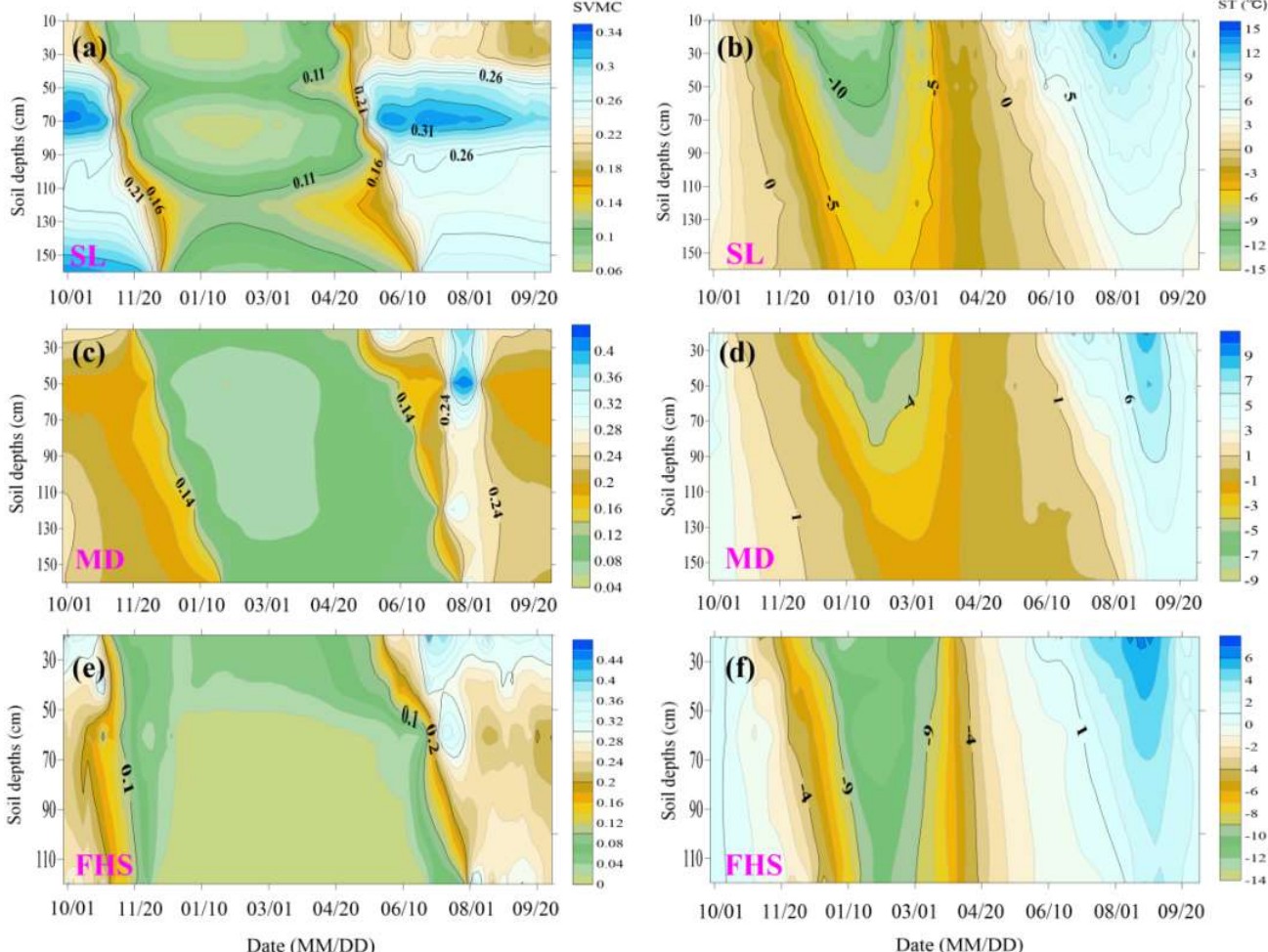

Figure 3: Vertical variations in soil temperature (ST) and soil volumetric moisture content (SVMC) in the different sites of the QTP.

Moreover, precipitation is a major water source that replenishes SG during the thawing stability period of AL according to the seasonal dynamic of SGL and precipitation (Fig. 2). The SGL in period D responded rapidly to rainfall and dropped to a low value in the absence of rainfall. In this period, AL reached its maximum depth in a year, while SGL was stable at a high groundwater level. For example, the SGL in Period D was stable above 80 cm depth at the SL station, rising rapidly with rainfall and slowly reducing to a stable level following rainfall (Fig. 2). The delayed SGL reduction after rainfall suggested that AL had a distinct role in water conservation and could significantly regulate soil water transport and runoff processes in a permafrost watershed. This water-conservation effect is crucial for alpine ecosystems.

## 4 Impact of ST and SVMC on SG in the study area

Figures 4 and 5 show the correlations between ST and SGL, as well as SVMC and SGL, in different soil layers at the SL station. The nonlinear relationships between the measured ST and SG were very good ($R^2 > 0.8$, $P < 0.01$) during period A (also the rapid freezing period of AL) (Fig. 4a). The results in Fig. 4a show a significant nonlinear correlation between SGL and ST at different depths ranging from 0–160 cm, with $R^2$ ranging from 0.81–0.98 ($P < 0.05$). However, significant differences were observed in the response of the SGL to the ST at different depths. For example, the range of ST at a depth of 10 cm at the SL station was approximately –10–5 °C, corresponding to rapid changes in SGL, while it narrowed to approximately –4–4 °C at a depth of 50 cm, –1–4 °C at a depth of 90 cm, and –0.2–3 °C at a depth of 160 cm (Fig. 4a). This is consistent with the gradual decrease in ST with increasing depth, which indicates that SG is more sensitive to ST changes in the deeper AL than in the shallow layer in period A. According to the fitting analysis shown in Fig. 4a, the correlation between ST and SGL gradually increased with increasing depth. In the early stage of period A, as the ST decreased, the soil froze gradually, and the liquid water in the soil transformed into solid ice, thereby decreasing SGL. The correlation coefficient between ST and SGL was the lowest in shallow soil, especially in the soil layer of 0–10 cm depth, indicating that the impact of shallow soil freezing on the SGL was weak during the freezing. This has a limited impact on the SGL, which can obtain some surface water through fissure infiltration and lateral flow.

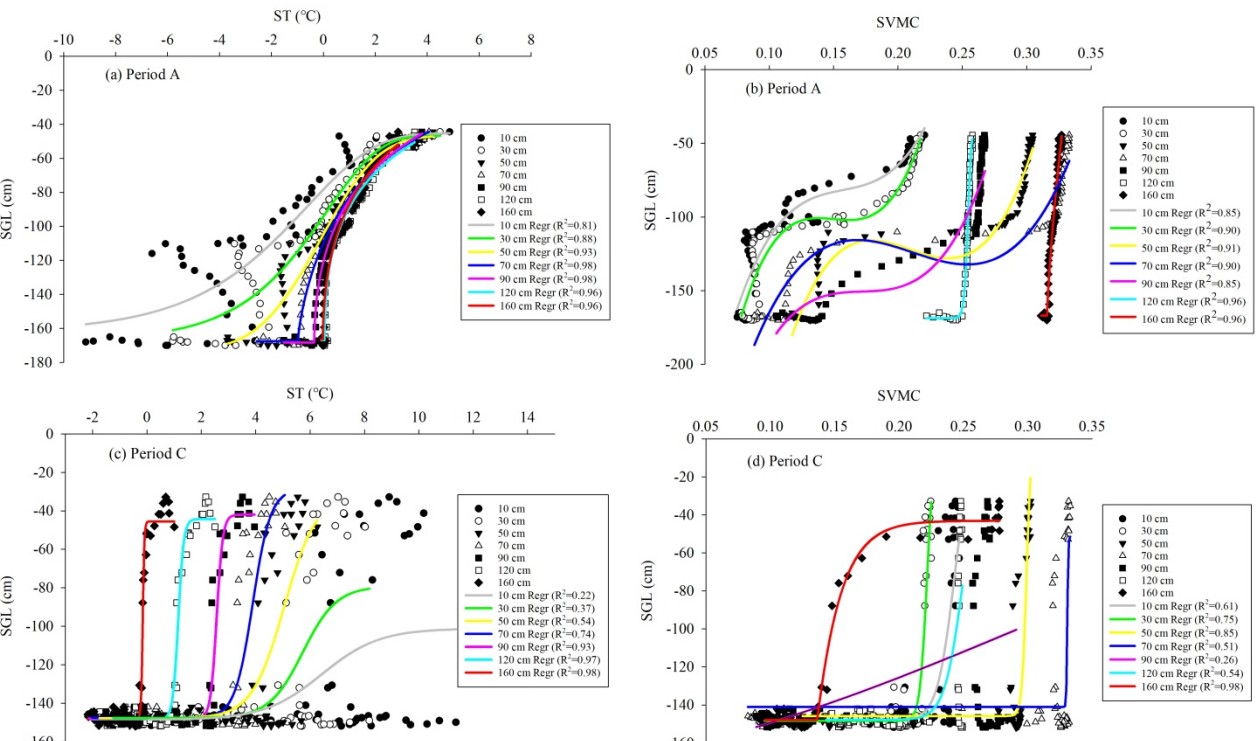

**Figure 4: Correlations of SGL and SVMC (ST) in different depths of AL during Periods A and C.**

Meanwhile, in period C (also the rapid thawing period of AL), the nonlinear relationships between the measured ST and SG fit well only in the deeper soil layers (below 50 cm depths) of AL (Fig. 4c). The SG in the deeper soil layers was more dominated by ST compared with that in the shallow soils. During this, there was no significant relationship between ST and SGL at depths of 10, 30, or 50 cm (P >0.05). ST at 70, 90, 120, and 160 cm depths had a significant effect on SGL (P <0.01); however, the overall correlation between ST and SGL improved with increasing depth, and the impact of ST on SGL became more significant (Fig. 4c). The shallow soils (0–50 cm) in AL rapidly thawed in the initial stage of period C, with rapidly rising SGL caused by the rapid supply of external water sources, including the potential lateral flow from thawed soils, rainfall and snowmelt infiltration (Yongjian et al., 2017). In the subsequent stage of Period C, soil water replenishment was relatively stable, and SGL dynamics were significantly affected by the thawing depths of AL dominated by ST, according to the good consistency between SGL and the thawing process (Figs. 2 and 3). In addition, the variation range of ST in different soil layers differed, as the ST range in shallow soil was substantially wider than that in deep soil (Fig. 4c). During the SGL variation period, the variation range of ST at a depth of 10 cm was 3–12 °C, and the response of SGL to ST in deep soil was more intense. For example, the fluctuation of SGL from low to high water levels was completed within the range of approximately –0.5–0.5 °C at 160 cm (Fig. 4c). This indicates that the ST impacts SGL dynamics only after soil thawing (>-0.5 °C).

Compared with ST, the impact of SVMC on SGL is limited during period A, although the nonlinear relationships between SGL and SVMC fit well in the 0–90 cm deep soils ($0.85 \leq R^2 \leq 0.96$, P <0.05) and 120–160 cm deep soils ($R^2 = 0.96$, P <0.01) (Fig. 4b). As ST decreases during period A, the liquid water in the AL soil gradually freezes into solid water, resulting in the observed SVMC (liquid water content) and the SGL accordingly decreases. Although there was a good relationship between SGL and SVMC at the 120–160 cm depth, the impact of soil moisture below 120 cm on SGL may be limited since SGL drops rapidly from high to low levels with minimal decrease in SVMC below 120 cm (Fig. 4b).

When the SGL changed in period A, the variation range of the SVMC first increased then decreased from the shallow to deep soil layers. The SVMC range gradually increased from 7–22% (10 cm) in the shallowest layer to 10–33% (70 cm) and gradually decreased to 32–33% (160 cm). This indicates that the response rate of the SGL to soil moisture first decreased and then increased with depth. By comparing the response rate of SGL to ST changes at different depths during period A (Fig. 4a), we found that deeper soils in the 0–70 cm soil layer showed a larger influence of ST on SGL. Although the SVMC and ST have good relationships with the SGL at a depth below 70 cm, the variation scope of the SVMC remains minimal. The freezing process of deep soil determines the uplift process of the AL lower boundary, which affects SGL. Therefore, the deep layer impacts SGL owing to the ST.

The SGL during period C was closely related to the changes in soil water of AL, especially in the shallow soil layers. As shown in Fig. 4d, the correlations between the SGL and SVMC also fitted well during Period C (RMSE <10). Except for the soil layers at 70–120 cm depths, where the correlation between the SGL and SVMC was poor ($R^2 < 0.6$, P >0.05), the different soil layers of AL showed a good nonlinear correlation between the SGL and SVMC (P <0.01) (Fig. 4d). The SGL

changes in soil layers at 0–50 cm depths were more correlated with SVMC, while SGL changes in a deeper AL layer (70–120 cm) were more significantly affected by ST, with better correlation (Fig. 4c, 4d). The correlation between SVMC and SGL was also very good ($R^2 = 0.93$) at the AL bottom (160 cm depth). The correlation patterns further confirmed that AL began to melt downwards from the surface soil during the warm season. Meanwhile, during the rapid thawing period of AL (period C), the shallow soil first melted and could receive water supply from rainfall and surface meltwater, resulting in the rapid response of the SGL. The SGL in a shallow soil layer has a high degree of response to SVMC change, while the thawing of deep soil in AL is primarily controlled by ST. A higher ST leads to deep soil thaws and the water flow channels between the surface AL and deep soil can be fully connected, causing alternations in SGL. When the deep soil thawing is not significant, the water-resisting effect of frozen soil becomes more prominent, and the formation of lateral flow and saturated soil flow to replenish the SG becomes challenging. The water movement of the thawed soil layer (including saturated soil flow) does not participate in SG dynamics, thereby weakening the response of SGL changes to SVMC than to ST in the 70–160 cm deep soil layer throughout period C. Therefore, the SG is primarily replenished by soil water in the shallow layers, and the SGL rise is significantly influenced by thawing depth, dominated by ST in deeper AL.

In period D (also the stable thawing period of AL), SGL was not sensitive to either ST or SVMC changes in different AL depths, according to the correlation analysis (Fig. 5). The range of ST fluctuations gradually decreases with increasing AL depth, though exceeding 2 ℃. The correlation between the ST at different AL depths and SGL during period D was very poor ($R^2 < 0.1$) (Fig. 5a). The SVMC showed an "increase-decrease-increase" process as depth increased. The correlation between SVMC and SGL in the shallow soil was better than that in the deep soil, with the highest correlation observed at a depth of 10 cm ($R^2 = 0.52$) (Fig. 5b). Indicating that during period D, SGL was more sensitive to SVMC changes in shallow AL (0–30 cm), whereas SGL in the remaining depths of AL was not sensitive to either ST or SVMC changes when SGL fluctuated at high values for an extended duration, while deep soil water remained predominantly or nearly saturated. According to the rainfall process during this period (Fig. 2), the dynamics of the SGL were primarily affected by rainfall, which could rapidly replenish the water storage in shallow soils and subsequently affect SGL.

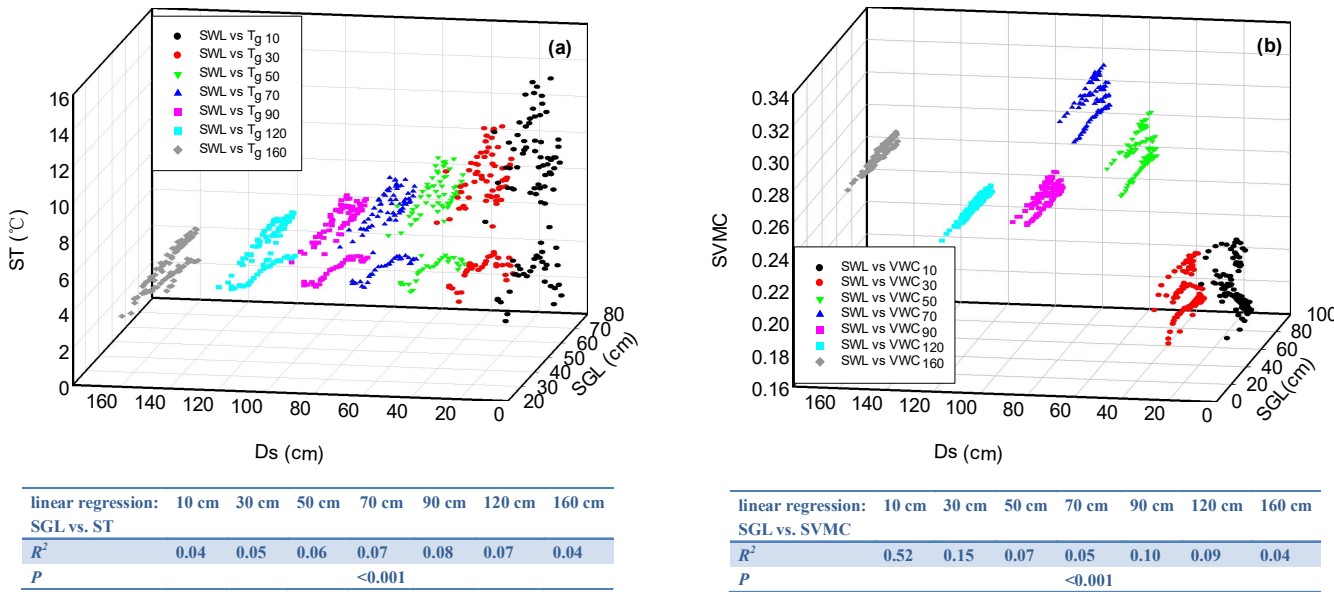

| linear regression: SGL vs. ST | 10 cm | 30 cm | 50 cm | 70 cm | 90 cm | 120 cm | 160 cm |
|---|---|---|---|---|---|---|---|
| $R^2$ | 0.04 | 0.05 | 0.06 | 0.07 | 0.08 | 0.07 | 0.04 |
| $P$ | | | | <0.001 | | | |

| linear regression: SGL vs. SVMC | 10 cm | 30 cm | 50 cm | 70 cm | 90 cm | 120 cm | 160 cm |
|---|---|---|---|---|---|---|---|
| $R^2$ | 0.52 | 0.15 | 0.07 | 0.05 | 0.10 | 0.09 | 0.04 |
| $P$ | | | | <0.001 | | | |

**Figure 5: Variations and correlations of SGL and ST (a) as well as SGL and SVMC (b) in different depths (Ds) of the permafrost active layer in Period D.**

## 5 Linkage between river discharge and SGL

To further analyze the impact of SG on river runoff, Fig. 6 shows the relationship between SGL and RL in the SL and FHS river basins during periods C and D, respectively. In both the SL and FHS watersheds, the SGL and RL processes were similar (Fig. 6a, 6c). As shown in Fig. 6a, SGL and RL showed stable fluctuations in the SL watershed during 15–25th of June (period C) and 19th of July to the 24th of August (early period D), with a high (P <0.05) "SGL-RL" process consistency. However, compared with early period D, the changes in the consistency of the SGL and RL processes in period C was slightly poor ($R^2$ = 0.07, P <0.05), which might have been caused by different AL thawing processes and the replenishment of snowmelt water to river runoff. In period D, the dynamic process of SGL was highly consistent with the regression and flood processes of river runoff, with a high correlation ($R^2$ =0.57, P <0.001). Therefore, the SGL contributed to approximately 57.0% of the RL changes in the SL watershed (Fig. 6b). Compared with period C, the consistency in period D was superior and the regression and flood processes were quicker.

Similar to the SL watershed, the SGL and RL in the FHS watershed had consistent fluctuations from the 31st of August to the 18th of October (period D), then showed a significant downward trend from the 2nd of October (late period D); their curves were also in good agreement (Fig. 6c). Figure 6d indicates that the SGL contributed approximately 65.8% of the RL changes in period D in the FHS watershed; however, the rate of decrease in SGL was larger than that of RL. The decrease in

SG was larger than 0.3 m, while that of RL was approximately 0.12 m. This may be because the surface soil begins to freeze as the temperature drops in late period D when the river water can be replenished by rainfall or snowmelt water. The freezing of shallow soil weakens the hydraulic connection between AL and the land surface, resulting in poorer water replenishment conditions, fewer water supply sources for AL, and a significant decrease in SGL.

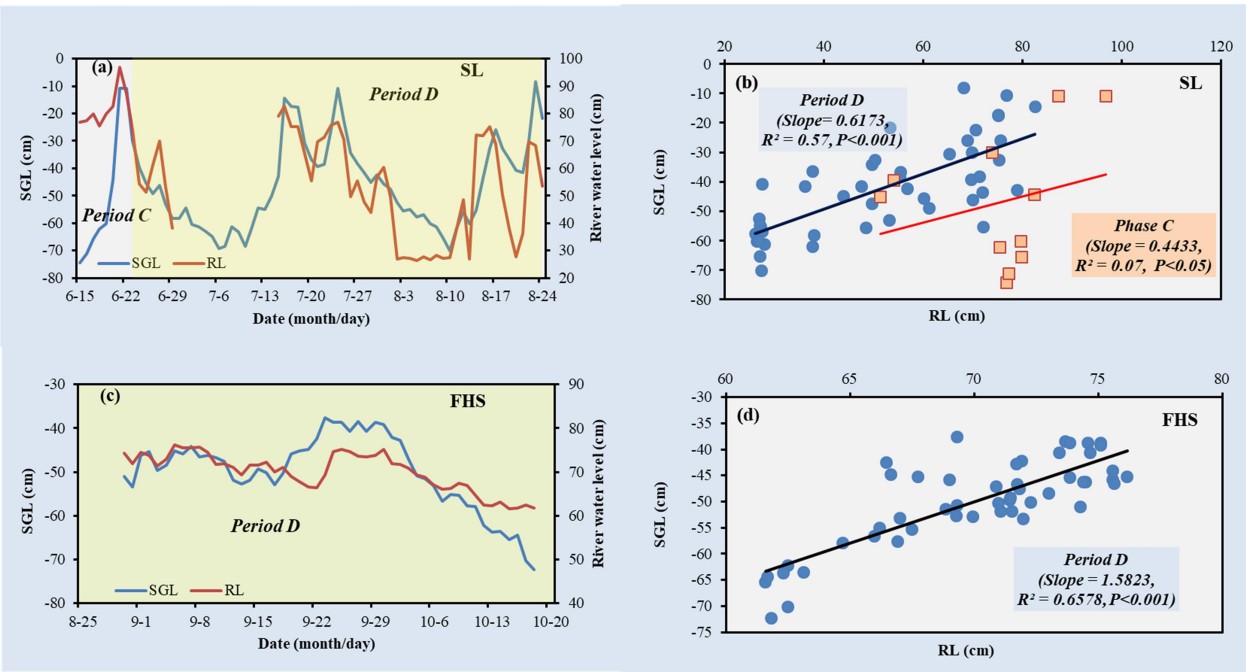

**Figure 6: Variations of SGL (a, c) and correlations between SGL and river level (RL) (b, d) in periods C and D, respectively, in the SL and FHS watersheds.**

## 6 Framework of watershed hydrology responding to the freeze–thaw of AL

According to the above analysis, the yearly hydrothermal changes have four distinct periods with seasonal AL freezing and thawing. The characteristics of the AL depth, SVMC, SGL, and surface RL changes in the four periods significantly varied (Fig. 7). In period A, ST rapidly decreased, causing the upper layer and bottom of the AL to freeze simultaneously, which limited the external water supply to the AL. The SGL rapidly decreased and reached the minimum value, as the surface RL rapidly decreased. However, due to the replenishment of rainfall runoff or snowmelt runoff, the rate of RL decline was relatively slower than that of SGL. In period B, the ST was consistently low, the AL was entirely frozen, and the SGL dropped to the lowest level with minimal changes, though the RL changes, in this period, were not always consistent with the SGL.

During snowmelt water replenishment, RL significantly increases, and the runoff process fluctuates and changes accordingly. In period C, the warming of the air temperature led to the rapid downward thawing of the AL from the surface, resulting in the rapid rise of SGL, reaching a peak value. Affected by the recharge of the lateral outflow of the SGL and rainfall runoff, the RL rose rapidly. In period D, high temperatures led to the deepest AL thawing, during which SGL and RL were significantly affected by rainfall. The SGL can increase rapidly with rainfall during the rainy season and maintain a higher value for a year with few fluctuations. In late period D, when rainfall decreases or is absent, the SGL rapidly falls and the river runoff is primarily replenished by a small amount of SG flowing out from AL, resulting in RL decreasing to a low value. When snowfall occurs in late autumn, snowmelt replenishes river runoff and causes the RL to rise. In addition, river runoff reverses the replenishment of the SG, leading to a moderate increase in SGL. Under the scenario of continuous warming in the future, AL will be thicker, the precipitation processes, and vegetation underlying surfaces will change leading to a more complex water regulatory mechanism associated with AL change.

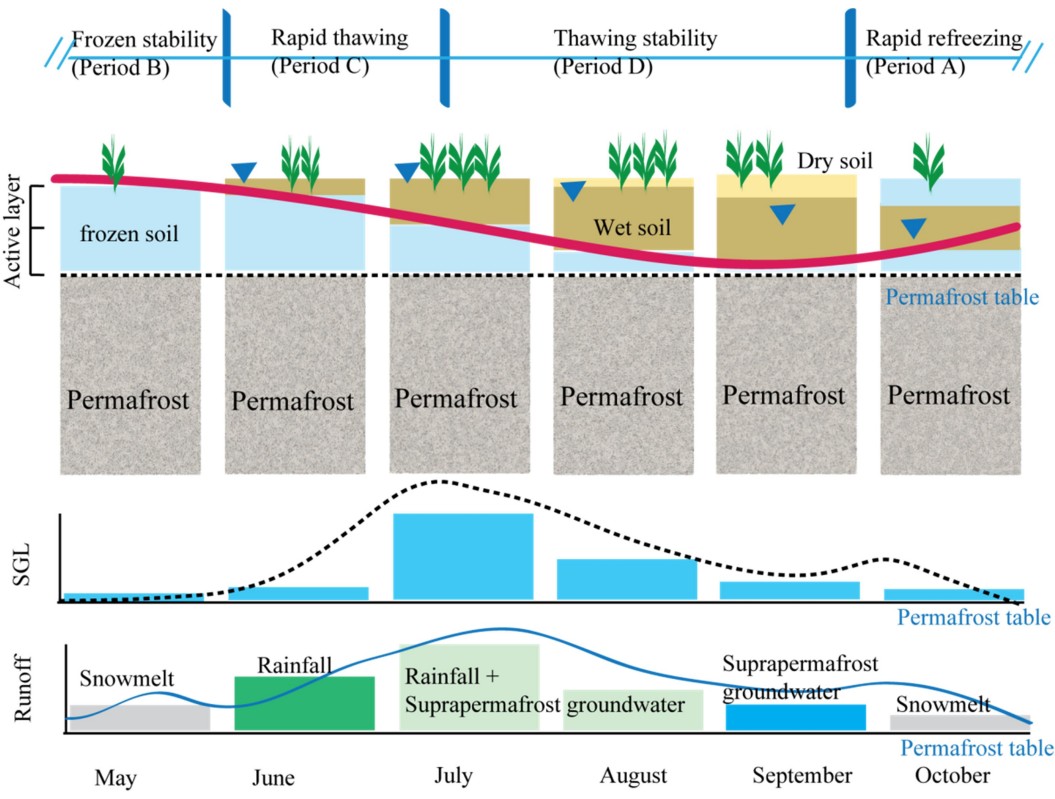

Figure 7: Framework of watershed hydrology responding to the freeze–thaw of the permafrost active layer.

**7 Discussion**

This study identified the tipping points of SG change during a hydrological year in different regions of the QTP via M-K mutation detection and divided SG data series into different stages. Although similar stages of seasonal SGL change were found in other permafrost regions of the Northern Hemisphere (Wellman et al., 2013; Koch, 2016; Tregubov et al., 2021; Koch et al., 2022), our study is the first to systematically summarize a four-stage pattern of SG seasonal change in a hydrological year. The seasonal variations of SG in the QTP are relatively consistent with the freeze-thaw characteristics of AL and mainly influenced by ST and SVMC changes, which has been previously concluded by studies of permafrost groundwater in Alaska (Hinkel and Nelson, 2003; Walvoord et al., 2012; Wellman et al., 2013), Canada (Woo and Xia, 1995; Clark et al., 2001; Liu et al., 2021) and other pan-Arctic watersheds (Young and Mingko, 2000; Dugan et al., 2009; Throckmorton et al., 2016; Koch, 2016; Koch et al., 2022). However, this study conducted a more detailed analysis of the ST and SVMC at different soil depths and variation stages impacting SGL changes and showed limited impact of SVMC on SGL, while the ST of AL had a larger impact on SG during period A. During period C, the SG in the deeper soil layers was more dominated by ST, while in the shallow soil layers of AL, it was closely related to the changes in soil water, primarily originating from meltwater of ground ice (Gubareva et al., 2018; Tregubov et al., 2021). In period D, SGL dynamic was not sensitive to either ST or SVMC changes at different AL depths and was primarily affected by rainfall. In this study, it was observed that SGL in period D was the highest compared with other stages of the year, which has been reported before (Rawlins, 2021; Tregubov et al., 2021). While this study further quantitatively revealed the contribution of SG to the RL changes in the thawing season of permafrost watersheds according to the correlation analysis. For example, over 50% of the surface runoff in period D was replenished by nearby SG in the SL watershed of the QTP. In addition, this study found differences in the degree of correlation between SG and surface water in different regions of the QTP based on the analysis of field observation data.

For example, the SG-RL hydrographs significantly differed in the SL and FHS watersheds (Fig. 6), indicating significant differences in the amount and rate of complementarity between SG and river runoff. The reasons for this difference may be attributed to the distance between the river and the measured point of SG, the slope, soil, and vegetation types (O'Connor et al., 2019). The farther the distance the longer the time and slower the process of mutual replenishment. The differences in water holding capacity and water resistance of different soils also directly affects the complementary amount between SG and river runoff (Wei et al., 2021). In addition, the thickening and deepening of the AL does not always exert a corresponding control on groundwater flows, and the impact that active layer thickening has on groundwater flows depends most on the position (depth) of the saturated thickness (O'Connor et al., 2019). This was further confirmed in our study. The SG observation points of FHS and SL are approximately 30 and 20 m far from the river, respectively, with the landscape type being alpine meadows (Renzheng et al., 2019; Qin et al., 2022). The saturated zone in the AL of the SL site was 50–80 cm in depth, while of the FHS site was only 0–30 cm in depth (Fig. 3). It was observed that the variation of SGL-

RL hydrographs was more similar in FHS than SL watersheds, which implies near surface saturated zone resulting in a closer complementary relationship between SG and river runoff (Fig. 6).

The large proportion of SG replenishing river runoff in this study implied that SG was a crucial replenishment source for permafrost basin runoff, especially during summer. While this has been previously suggested (Bense et al., 2009; Shepelev and Pavlova, 2014; Fischer et al., 2017), our analysis provides a quantitative assessment of this impact and confirms it in different regions of the QTP. According to $R^2$ in period D, 57.0–65.8% of the river runoff was replenished by SG in the study areas. In contrast, the SG contributed only approximately 10.0% of the RL changes in period C. Tregubov et al. (2021) reached a similar conclusion in the Ugol'naya-Dionisiya River, Northeast Russia, and reported that the contribution of SG to river discharge in period C was 10–30% of the total discharge. The difference of contribution rate in periods C and D was closely related to the storage capacity of suprapermafrost reservoir and the recharge intensity of surface water to SG (Ma et al., 2017). During period C, when the AL began to thaw and the storage capacity of suprapermafrost reservoir was small, the replenishment of SG to river runoff was limited (Qin et al., 2022). During summer, the seasonal thaw moved downward and thus the storage capacity of suprapermafrost reservoir increased, leading to most of the infiltrated rainwater being stored in the AL (Bosson et al., 2013). The larger recharge intensity of surface water (rainfall) and storage capacity of suprapermafrost reservoir result in the drainage of SG to river runoff in period D.

Our data showed two distinct high-water soil layers (saturated zone) in the AL of the QTP during summer: a near surface high-water zone (0–90 cm depth) and a deep layer (from 110 cm depth to the bottom of the AL), with a dry layer (90–110 cm depth) generally present in the middle AL (Fig. 4). These results differ from those of previous studies on the Arctic watersheds (Quinton, 1997; Street et al., 2016; Sebastian et al., 2023), having only one saturated zone in the AL of different specific regions during thawing season, e.g., a near surface saturated zone in the riparian zone and deep saturated zone in the hillslopes (O'Connor et al., 2019). The feature could be related to rainfall characteristics, Topography (slope), vegetation root depth, as well as water-holding capacity, soil type, and composition at different layers (O'Connor et al., 2019). Abundant summer rainfall could vertically infiltrate the near-surface soil layer and increase the moisture content of shallow soils (even saturated) in the QTP (Fig. 2) and Arctic watersheds (O'Connor et al., 2019; Sebastian et al., 2023). Figures 2 and 3 show that the surface rainfall could infiltrate downward into a maximum 90 cm depth in the AL on the QTP during summer rainfall events. The soil in the middle AL often has low moisture content limited by the weak infiltration capacity and poor water-holding capacity (Qin et al., 2022), while potential gravity and fissure waters from surface water and shallow soils, coupled with the water-resisting effect of the permafrost table leads to highly saturated zone in the bottom of the AL. In addition, the maximum AL depths of several Arctic watersheds are <100 cm (Hinkel and Nelson, 2003), while it was 1.0–2.0 m in the QTP. This and the differences in soil texture and hydrological characteristics could partly explain the two observed saturated zones in the AL of QTP. The two saturated zones in AL, in turn, lead to two potential water sources of lateral flow replenishing river runoff and permafrost lakes, which differs from the conclusions of only one major lateral replenishment zone in AL studied in Arctic watersheds (Koch, 2016; O'Connor et al., 2019; Manasypov et al., 2020). These findings provide novel insights into hydrological analysis and simulation in permafrost regions.

## 8 Conclusions

This study analyzed the seasonal dynamics of SG and the correlations between SGL, ST, SVMC, and RL at different stations in the QTP. The variation process of the SGL during the hydrological year can be distinctly divided into four periods, namely a rapid falling period (October–middle November) (A), a stable low-water period (late November–May) (B), a rapid rising period (approximately June) (C), and a stable high-water period (July–September) (D). This synchronously corresponds to the ST and SVMC variations in AL, which experienced a rapid freezing period, frozen stability period, rapid

thawing period, and thawing stability period during periods A–D of the SGL, respectively.

     ST and SVMC in AL significantly influence the SGL changes in permafrost watersheds, however, varied in the four periods. Compared with the other periods, the SGL in period D was permanently higher. The correlations between SGL and ST, as well as SGL and SVMC, were relatively poor, and the SGL change responded well to rainfall. During periods A and C, SGL had a good nonlinear correlation with ST and SVMC in the AL, while the correlations varied at different depths.

During period C, when AL rapidly melted, SVMC in the shallow soils (0–50 cm depth) better correlated with SGL, whereas ST in deeper AL soil (below 50 cm) showed a closer association with SGL than with SVMC. Therefore, SG in period C was primarily replenished by soil water in shallow layers, while the SGL rise was significantly influenced by the thawing depth dominated by ST in deeper AL. In period A, there is a significant nonlinear correlation between SGL and ST at different AL depths ($0.81 \leq R^2 \leq 0.98$, $P < 0.05$). As the depth increases in the AL, the impact of the SVMC on the SGL weakens, whereas

ST gradually becomes the dominant factor affecting the SGL.

     According to comparative analysis, the retreat and flood processes of river runoff were consistent with the SGL changes in periods C and D, the primary annual runoff periods in permafrost basins. The RL dynamics were closely related to the SGL changes during these two periods. The SG and river runoff were interconnected, and their water linkages were significantly affected by the freeze–thaw state of AL. The SGL contributed approximately 10.0% of the RL changes in

period C, whereas in period D approximately 57.0–65.8% of the surface runoff in permafrost watersheds was replenished by SG, primarily via rainfall infiltration. The SG is a crucial and potential water source for alpine permafrost watersheds.

     In conclusion, the characteristics of SG vary at different periods of the year and have a crucial regulatory effect on the hydrology of permafrost watersheds. Continuous climate warming will thicken AL, and alter potential precipitation and alpine vegetation. Furthermore, the change mechanism underlying SGL will correspondingly become more complex and

require further research.

**Author contribution**

Jia Qin and Yongjian Ding developed the idea and outlines of the article. Jia Qin prepared the manuscript with contributions from all co-authors.

**Competing interests**

The authors declare that they have no conflict of interest.

**Acknowledgments**

This work was supported by the National Key R&D Program of China (Grant No. 2021YFC3201102-02), the National Natural Science Foundation of China (Grant No. 42171028, 42330512, 42371152 and 41877156), the State Key Laboratory of Frozen Soil Engineering Foundation (Grant No. SKLFSE202110), and the Open Project of the State Key Laboratory of

Cryospheric Science, China (Grant No. SKLCS-OP-2020-7).

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
