# Peer review of "Links between seasonal suprapermafrost groundwater, the hydrothermal change of the active layer, and river runoff in alpine permafrost watersheds"

_EGUsphere, 2023_

## Author Response (AR1)

**Responses to reviewer 1**

*Thank you very much for the comments and suggestions, which could help to significantly enhance the article level. All our authors have discussed the review comments and carefully revised each item based on the opinions.*

*Reply to the following comments one by one:*

- The research question is not clear. In Introduction, authors invested quite some effort in showing the importance of supra-permafrost groundwater (SG), but much less in reviewing the status quo of SG dynamics and proposing specific scientific research questions. Such issues make the reader very confusing at the first glance of this paper. For instance, in lines 58-67, the authors listed the influencing factors of SG variation, but without presenting how these factors impact SG variations and what questions need to be solved. One sentence of "However, there is a shortfall in systematically revealing the linkage between the seasonal hydrothermal change of AL, SG, and surface runoff." is far from enough.

  *Reply: thanks a lot for the reviewer's comments and suggestions. We all authors agree with the reviewer's opinion and have carefully revised the "Introduction". In this section, the detailed research dynamic of SG and the corresponding references have been added in the text. In addition, we have adjusted the structure and revised corresponding sentences to highlight the research questions. All the revisions have been marked in yellow.*

- The structure of the manuscript needs a major overhaul as it is difficult to find a red line for each section or paragraph. For example, why is the Sigmoid–Boltzmann formula in Result? It should be in Methods and materials 2.4, and why this formula is used for analysis? What is the key point in "3 Seasonal characteristics of SG"? Can you use one brief sentence at the beginning of each paragraph to give some general findings or ideas? Meanwhile, I may suggest change the old 3 into "Seasonal characteristics of SGL, ST, and SVMC", and move related descriptions from 4 to 3. What's more, the current Discussion is not a not real "discussion", the authors showed the importance of SG again, but without presenting what's new in this study compared with previous studies. Moreover, I do think the "framework" should be in "discussion" with a focus on what general or/and new patterns you have found, what processes are involved, and corresponding implications for future studies.

  *Reply: We agree with the opinion and suggestion. We have adjusted the structure of the manuscript. The section 3 has been revised to "Seasonal characteristics of SGL, ST and SVMC". The corresponding text in section 3 and*

*4 also has been revised. One brief sentence has been added at the beginning of each paragraph in section 3 and 4 to give some general findings (in yellow). The formulas selected in the text are based on the results of fitting analysis of non-linear least square method according to existing reference (Wang et al., 2012). We have moved the formulas and instructions in section 4 to section 2.4, so as to make the structure more reasonable. In addition, we have rewritten the "Discussion" considering the reviewer's comments (in yellow).*

- More details are needed in Methods and materials to make sure the reliability of all results in this study. For example, (i) how to define the four different periods of SGL? What do you mean "tip points"? How to select the tip points in a scientific way? (ii) more detailed calibration procedures for SGL, SVMC, and ST sensors are also needed; (iii) how are the geographical distribution of observations for SGL, SVMC, and ST, meteorological variables and river level? SGL, ST, SVMC data were at point-scale, while the river level reflected the overall discharge at the catchment scale. How do authors consider scaling issues throughout the analysis? (iii) what is the hydrogeological condition in the study area? Is SG recharged by river or river is recharged by SG? Authors need to make it very clear. Otherwise, most readers will not be able to assess the results as well as the reasoning behind the interpretation.

***Reply:** thanks for the detailed comments and suggestions. (i) For a detailed study on seasonal SGL, the tip points of SGL data series were analyzed using the Pettitt test (Pettitt, 1979). According to the turning trends and tipping points, SG hydrographs of a hydrological year were divided into four periods. We have added the corresponding methods and explanations in the section 2.4 (line 208-209, marked in yellow) and section 3 (line223-229, marked in yellow).*

*(ii) the text about detailed calibration procedures for SGL, SVMC, and ST sensors has been added in the third paragraph of section 2.3 (in yellow) .*

*(iii) We have revised the Figure 1, in which the geographical distributions of observations for SGL, SVMC, and ST, meteorological variables and river level have been added.*

*In addition, some sentences were added in the section 2.1 (line 154-155) and section 2.3 (line 203-206) to explain the scaling issues about SG and RL. The selected watersheds are commonly located in relatively open terrain with wide and shallow valleys. the experimental watersheds are relatively small (as shown in Fig.1) The typical alpine hillslopes were selected in the central part of the experimental watersheds (SL and FHS) where the groundwater flow field on the hillslopes was cut by the river (with ground ice exposed at the edge of the riverbed and with an obvious exchange between SG and river runoff), to observe the "SGL-RL" linkage during the AL thawing period.*

*(IV) we have added two paragraphs in section 2.1 (lines 116-125, and lines 149-163 ) to illustrate the hydrogeological condition in the study area (marked in yellow).*

Four different stations with climatic gradients are selected in this study, a comparative analysis is thus expected for readers. However, almost nothing related is shown in this paper.

*Reply: thanks for the comment. The analysis in four different stations selected in this study was aim to shown common features of the linkage between SG, ST, SVMC, and RL in alpine permafrost watersheds. So, the manuscript has weakened the difference. In the new revised "Discussion", we have added some comparative analysis about the different stations as well as the other regions in the permafrost regions of the world.*

- More in-depth analyses are needed to illustrate the impact of ST and SVMC changes on SGL variations, the role of SG in regulating river runoff, and how SGL responses to rainfall events. Currently, most analyses are simple descriptions about the statistical relationship between variables.

   *Reply: thanks a lot. In the revision manuscript, we have considered the suggestions, and added more analyses to illustrate the impact of ST, SVMC and rainfall on SGL, and the linkages of SG-RL in the section 3, 4 and 5, as well as the "Discussion" section.*

- Pay attention to the consistency problems in this paper. For example, in lines 95-96, it says "the analysis was conducted in a hydrological year (from October 1 to September 30)", while the time series in Figure 3 are contrasted with this, and also different with each other for each small figure. Meanwhile, such time series in Figure 3 are also different to those in Figure 2, making it hard for readers to follow expressions in line 132.

   *Reply: thanks and we have revised the figure 2, 3 and corresponding text throughout the manuscript.*

- The level of English does not yet meet the standard required for scientific publications and requires a detailed round of proof reading by a native English speaker with hydrology background.

   *Reply: thanks. The English of the manuscript has revised by expert English editors of "Editage". To enhance the English, we have asked other English editor with hydrology background check and revised the proof again. All the revisions were marked in the text.*

Specific points:

L.12: what is runoff concentration here?

 *Reply: the original meaning of the words in the sentence was to refer the "runoff confluence". We have revised it (line 12, in yellow).*

L.38: change into "it plays a crucial role in regulating land surface processes…"

 *Reply: ok, we have revised it (lines 42-43, in yellow).*

L.39: what do you mean "the SG maintains a high value", I suppose it should be the SGL?

 *Reply: Thanks and we have replaced SG by SGL (line 43, in yellow). And some other similar expression was also were revised in the manuscript.*

A separated table with site-specific characteristics including location, altitude, vegetation type, annual mean precipitation, air temperature, soil properties, and so on, could better illustrate the gradient.

 *Reply: Thanks for the suggestion, and we have added a separated table (table 1) including the information of vegetation type, annual mean precipitation, air temperature, and maximum depths of AL. The information of location and altitude of the four watersheds were added in Figure 1.*

L.101: The writing should be more concise. Such expressions like "nearby national weather stations" without the distance are not reliable.

 *Reply: Thanks for the suggestion and we have checked the corresponding expression in the text. The sentence "--nearby national weather stations" also has been revised (lines 171-173)*

L.121-122: Refs are needed here for the method and algorithm.

 *Reply: ok, we have added the Refs of the Levenberg–Marquardt method and Universal Global Optimization algorithm in the revised manuscript (lines 213-214).*

L.126: change into the seasonal variation of SGL…

 *Reply: Thanks, this sentence was deleted in the new revised manuscript.*

L.163-164: refs are needed to support your inferences.

 *Reply: ok, the reference has been added in the sentence (lines 289-290).*

L.233: use correlation analysis results to prove this.

 *Reply: ok, we have added the significance value of correlation in the sentence (line 357).*

L.299-300: refs are needed again.

 *Reply: Thanks, we have rewritten the "discussion", and the original sentence has been deleted.*

L.317: only variables of ST and SVMC are considered for evaluating their effect on SGL changes, how could you get the conclusion that they are the primary impact factors.

 *Reply: thanks. We have checked the expression and revised the corresponding sentence to make it more accurate (lines 486-487).*

**Responses to reviewer 2**

The manuscript "Links between seasonal suprapermafrost groundwater, the hydrothermal change of the active layer, and river runoff in alpine permafrost watersheds" has systematically analyzed the impacts of freeze–thaw processes of active layer on seasonal suprapermafrost groundwater (SGL), and the links between SGL and surface runoff based on the field observations. The framework of watershed hydrology responding to the freeze-thaw of the permafrost active layer also was explored. The topic in interesting and this manuscript is significant in hydrological science in clod regions. The methods and the study conclusion are convincing. In my opinion, the manuscript is suitable for publishing in the HESS after some minor revisions concerning as follows:

1. The English of the manuscript is good, while there are a few sentences which are too long and a bit complex. It is suggested to be separated in a few short sentences. For example, the sentence in line 214-217 "Although the SVMC and ST both have good relationships with the SGL below a depth of 70 cm, the variation scope of the SVMC is minimal, and the freezing process of deep soil determines the uplift process of the AL lower boundary, which affects the SGL. Therefore, the deep layer also more directly impacts the SGL owing to the ST."

   *Reply: Thanks. We all authors agree with the opinion, and have revised the sentences. In addition, we have done a detailed round of proof reading by a native English speaker with hydrology background. (All revision could be tracked in the revision manuscript)*

2. The font of text and label in some Figures, such as the Figure 1 and Figure 4, are too small and not very clear. The color of the lines in the subtable of Figure 5b is inconsistent.

   *Reply: Thank for the opinion and we have accordingly revised the figures, including Figure 1, 4, and 5.*

3. The section "Discussion". In the section, I recommend to add some discussions about the different impacts of vegetation and the slope on SGL dynamics, although they maybe not affect the conclusion of the study.

   *Reply: thanks. We have rewritten the "Discussion" according to two reviewers' suggestions.*

4. Line 337, one more space in "The change----".

   *Reply: thanks, we have deleted the space in the text (line 506).*

5. Line 58 and Line 62, the SG should be SGL?

   *Reply: that is right, and the SG has been replaced by SGL (lines 64 and 69).*

6. In line 96, the "---in a hydrological year". Does it refer a specific year or the annual average value?

   *Reply: it means a whole hydrological year (from October 1 to September 30). According to reviewer's comments, we have checked the potential unclear expressions in the manuscript. And the data series in subfigures of Figure 3 have been revised to be consistent with each other.*

7. Why the data of station HLG was not shown in Figure 3? No data or other reasons? It should be introduced in the text.

   *Reply: Some data of ST and SVMC in the HLG station is missing. The corresponding instructions have been added in the first paragraph of section 2.3 (lines 183-185).*

---

## Referee Report (RR1)

The manuscript "Links between seasonal supra-permafrost groundwater, the hydrothermal change of the active layer, and river runoff in alpine permafrost watersheds" has been thoroughly revised by the authors according to reviewers' comments and suggestions. I thus suggest that it can be considered for publication after technical correction.

---

## Author Response (AR2)

Review comments:Please supply the detailed locations (longitude and latitude) of the four permafrost stations.

**Reply:** *Thanks for the comment and suggestion. The detailed locations (longitude and latitude) of the four permafrost experimental watersheds have been added in Table 1.*